# Temperament, Plasticity, and Emotions in Defensive Behaviour of Paca (Mammalia, Hystricognatha)

**DOI:** 10.3390/ani11020293

**Published:** 2021-01-24

**Authors:** Selene S. C. Nogueira, Sérgio L. G. Nogueira-Filho, José M. B. Duarte, Michael Mendl

**Affiliations:** 1Laboratório de Etologia Aplicada, Universidade Estadual de Santa Cruz, Av. Jorge Amado, km 16-Salobrinho-Ilhéus-BA, Ilhéus, BA 4566-2900, Brazil; slgnogue@uesc.br; 2Núcleo de Pesquisa e Conservação de Cervídeos (NUPECCE), Universidade Estadual Paulista (UNESP), Jaboticabal, SP 14884-900, Brazil; mauricio.barbanti@unesp.br; 3Centre for Behavioural Biology, School of Veterinary Science, University of Bristol, Langford House BS40 5DU, UK; mike.mendl@bristol.ac.uk

**Keywords:** behaviour, cognitive bias, escape behaviour, personality, stress, temperament

## Abstract

**Simple Summary:**

The paca (*Cuniculus paca*), a Neotropical caviomorph rodent, provides the most sought-after game meat in all its range, and it therefore faces high hunting pressure and consequent poor welfare. The species is categorised as having a conservation status of “least concern” and appears resilient to over-hunting by humans, which may be related to individuals’ behavioural characteristics. To investigate this, we submitted captive pacas to temperament (personality) tests designed to assess individual responses to short challenges and to evaluate individuals’ emotional states. Our results showed that paca with a “restless” temperament performed more abnormal behaviour and less exploratory behaviour in a test of defensive behaviour, which elevations in faecal glucocorticoid metabolites indicated to be stressful. Plasticity in defensive behaviour was inferred from changes in behavioural responses and apparently rapid adaptation to different levels of risk. Our results suggest that individual differences and consistency of behavioural responses displayed by paca toward challenges may reflect a generally flexible and successful defensive behavioural response that underpins the paca’s survival, despite the threat of overhunting throughout its range.

**Abstract:**

Within a species, some individuals are better able to cope with threatening environments than others. Paca (*Cuniculus paca*) appear resilient to over-hunting by humans, which may be related to the behavioural plasticity shown by this species. To investigate this, we submitted captive pacas to temperament tests designed to assess individual responses to short challenges and judgement bias tests (JBT) to evaluate individuals’ affective states. Results indicated across-time and context stability in closely correlated “agitated”, “fearful” and “tense” responses; this temperament dimension was labelled “restless”. Individual “restless” scores predicted responses to novelty, although not to simulated chasing and capture by humans in a separate modified defence test battery (MDTB). Restless animals were more likely to show a greater proportion of positive responses to an ambiguous cue during JBT after the MDTB. Plasticity in defensive behaviour was inferred from changes in behavioural responses and apparently rapid adaptation to challenge in the different phases of the MDTB. The results indicate that both temperament and behavioural plasticity may play a role in influencing paca responses to risky situations. Therefore, our study highlights the importance of understanding the role of individual temperament traits and behavioural plasticity in order to better interpret the animals’ conservation status and vulnerabilities.

## 1. Introduction

The ability to successfully avoid predation is critical to individual survival and can be influenced by variation in behavioural characteristics [1,2] including anti-predator vigilance [3], aggression levels [4], and predator detection ability [5]. Individuals may thus vary in their boldness and risk-taking in the face of challenge due to underlying differences in vigilance, aggression, and other capabilities including the speed of decision-making [6]. The ability to adapt flexibly to changes in the environment is also likely to influence how well an individual deals with challenges, and to underpin why some species or individuals are able to thrive and persist in threatening environments better than others [7].

The paca (*Cuniculus paca*), a Neotropical caviomorph rodent, provides the most sought-after game meat in all its range [8], and therefore faces high hunting pressure [9,10,11]. The unsustainable hunting of the paca has led to local depletion of the species in several locations, especially due to its relatively low reproductive rate [12,13,14,15,16]. However, due to its wide distribution, occurrence in a number of protected areas, and presumed large population, the paca is unlikely to be declining and, consequently, it is categorized as being of least concern on the IUCN Red List [17]. Indeed, van Vliet and Nasi [18] suggest that the paca’s behavioural plasticity traits are important contributors to its resilience to hunting.

The species is able to thrive in a wide variety of environmental conditions [19]. Generally described as having a nocturnal habit [20], the paca may present crepuscular and dawn activity [21,22], and in captivity it is easily conditioned to diurnal habits [23]. Free-ranging pacas usually live alone or in pairs [24], showing a high level of aggressiveness against conspecifics of the same sex when defending their territory [24]. It has been suggested that the solitary behaviour mostly reported for paca is an adaptive response to high hunting pressure, because free-ranging pacas have been seen in groups in areas with low hunting pressure [10,25]. This is corroborated by the vocal repertoire of the paca, which comprises eight calls [26], suggesting the potential for complexity of communication and hence a more social nature as observed in other social caviomorph species living in large groups, such as the capybara (*Hydrochoerus hydrochaeris*) [27]. Additionally, the paca shows capacity to live in high population densities (up to 96 individuals per km^2^ [19]) and has burrowing habits that are related to group-living [28,29]; farmers usually breed this species in groups (one male with three to five females) [23]. 

Whatever its true social nature, a flexible response to challenging situations may be linked to the paca’s resilience to hunting pressure, and variation in boldness or risk-taking can have a strong influence on individual defensive responses and resulting survival. Thus, studying the paca’s behaviour will allow us to understand its responses to predation/hunting risks, and to foster proper management for the sustainable use of this common and non-threatened species used as a source of food in Neotropical forests. Here, we sought to investigate whether we could detect consistency across time and situation in the responses of individual pacas to brief challenging scenarios, and whether behaviour shown in these temperament tests predicted individual responses to longer and more threatening situations, including simulations of chasing and forced contact with a human “predator”. 

We were also interested in whether characteristics of stability and plasticity, conceptualised as core components of a “big two” human personality theory [30], could be detected in paca. This theory states that some behavioural characteristics are more associated with stability (e.g., disposition toward coping with stress and negative emotions, acting cautiously, controlling impulses), while others are more associated with plasticity (e.g., disposition toward exploration, flexibility, and tendency to have positive emotions). Considering this idea, we would expect individual pacas to show some stability in their responses toward an environmental threat, but also to be able to alter some of their responses according to the risk levels of these threats, thus showing flexibility (plasticity). 

To investigate these issues, we measured the responses of captive pacas to three short challenge tests to assess indicators of their temperament. Additionally, we analysed their defensive responses using the modified defence test battery (MDTB [31]) and investigated links between the emerging temperament measures and response to simulated hunting. We also measured faecal glucocorticoid metabolites prior to and following the MDTB, to determine whether temperament measures were associated with physiological stress responses to the test events [32,33,34]. Moreover, we trained pacas on a judgement bias test (JBT) [35,36], which involves assessing responses to ambiguous stimuli that predict potentially positive or negative outcomes. This test has generally been used as a marker of animal affective states following the hypothesis, based on human psychology studies (e.g., [37,38]), that individuals in a more negative state will be more likely to treat ambiguous stimuli “pessimistically” as predicting a negative outcome. Here, we also measured speed of learning of the discrimination task on which the JBT is based as an indication of behavioural plasticity, as well as responses to ambiguous stimuli during JBTs carried out before and after exposure to the MDTB.

We predicted that we would find stable variation in our temperament measures amongst pacas, as observed in other captive wild mammals (e.g., *Pecari tajacu* and *Tayassu pecari* [39]). We also predicted that individuals that were bolder and less disturbed in challenge tests would also be less affected by the MDTB, show a smaller physiological stress response, and make fewer “pessimistic” decisions in the JBT, because links between behavioural and physiological responses to challenge have been observed in other rodents [40,41,42]. We also expected to see plasticity of responding during the duration of the MDTB, because pacas exhibit behavioural flexibility [18,43].

## 2. Materials and Methods 

### 2.1. Ethical Note

This work followed the “Principles of Laboratory Animal Care” (NIH publication No. 86-23, revised 1985) and was approved by the Committee of Ethics for Animal Use (CEUA) at the Universidade Estadual de Santa Cruz (proc. #021/16). 

### 2.2. Study Site, Animals, and Sequence of Experimental Procedures

The study was conducted at the Applied Ethology Laboratory at the Universidade Estadual de Santa Cruz, Bahia, Brazil (14°47′39.8″ S, 39°10′27.7″ W). The animals, eight males of the species *Cuniculus paca*, were all 1.2-year-old adults, born and bred in captivity from different parents, weighing on average 6.5 (±0.5) kg. We did not use females in case there was an influence of hormone changes across the ovulatory cycle on behavioural or physiological data which might mask effects, because the whole study took over 180 days to measure all behavioural responses. We acknowledge that our findings may thus not be generalisable across sexes.

The animals were individually housed in 11.3 m^2^ pens (Figure 1A), covered with a chain-link wire screen. Each home pen was divided into two sections: one covered area of 3.0 m^2^ (2.0 m length × 1.5 m width) and an additional area, comprising a partially sheltered section and a “solarium” section with a cement floor which allowed unobstructed exposure to natural sunlight. The walls between pens consisted of 1.5 m-high wire fencing to which were fixed zinc plates to avoid visual contact with the neighbouring pens. Following Smythe [44], we conditioned the animals for daytime activity by feeding them during the day to alter their natural nocturnal habits. This procedure is adopted by paca farmers [25], and animals easily adapt to this change, interacting with their keeper during the daytime. The pacas were fed a diet composed of 80 g of rabbit pellet diet and 100 g of banana (*Musa* sp.), 120 g of sweet potato (*Ipomoea batatas*) and 200 g of mango (*Mangifera indica*) supplied once a day at 08:00 h, according to the feeding regime described by Aldrigui et al. [45]. Water was supplied ad libitum in a bucket, fixed 0.1 m high on a wall of the pen to prevent paca defecating inside it. 

For judgement bias training and both judgement bias tests, each animal was transferred to a cage (1.5 m length × 0.7 m width × 0.8 m height) made of metal with a wooden grid floor. These cages were previously used to study the nutrition of paca, and animals were observed to exhibit normal species behaviour in them [45,46]. The cages were located outdoor side by side, with a distance of 1.0 m between them. These cages were protected from precipitation by an overhanging tile roof, with its longitudinal axis positioned in an east–west orientation. Each cage had a feeder, a drinking trough, and a shelter. Although paca have poor daytime vision [24], they occasionally show crepuscular and dawn activity [21,22] and, as mentioned before, usually interact with their keeper during the daytime; thus, we set up a plastic screen (mesh size 12 millimetres) in front of the cages to prevent visual contact between animals and researchers. For each cage, we made a hole (25 cm^2^) in the plastic screen to provide rewards when appropriate (see below). The animals remained in these cages (approximately 30 min per session) just to perform the judgement bias training and judgement bias tests, and at the end all individuals were returned to their individual housed pens. 

To evaluate individuals’ defensive behaviour (details below), a test arena was built, following the example in ref [47]. The arena (Figure 1B) had a corridor shape on a dirt floor with only a few occasional clumps of grass; measured 15.0 m in length and 4.0 m in width; and was surrounded by a chain-link fence 1.5 m high to which zinc plates were fixed to both prevent visual contact with the outside environment as well as the animals’ escape. At the centre of this enclosure, there was a chain-link wall 1.5 m high dividing the test arena into two parallel corridors measuring 2.0 m in width each. At both ends of this central wall, there was an opening, 1.0 m wide, which allowed the paca to move throughout the entire arena, unless guillotine doors were closed to block these openings as in the forced contact test (described below). Zinc plates were fixed to the central wall to avoid visual contact between corridors during tests. A digital camcorder (JVC, model GZHD500; Tokyo, Japan), fixed on a tripod 1.6 m high above one extremity of the test arena, was used to continuously video-record the animals’ behaviour during all four tests. 

This study was conducted during a period of 180 days following the timeline sequence presented in Figure 2. The timeline was designed to evaluate the individuals’ behavioural responses before and after the modified defence test battery—MDTB. Initially, we collected faecal samples to determine basal faecal glucocorticoid metabolite concentration before any interference with the animals (no stress induced) and at the end of the experiment to evaluate the increase in glucocorticoid metabolite concentration after the modified defence test battery—MDTB (after stress induced). To assess temperament across time and contexts, we carried out tests in different situations and separated by at least 60 days. To evaluate affective states, we used a judgement bias test (JBT) which required initial training of animals to respond to cues by showing “go” or “no-go” responses, followed by testing on ambiguous cues before and after defensive behaviour to evaluate the effects on animals of stress induced by the modified defence test battery (see below). During both sets of faecal sampling, novel object test (ball) and predator-like model, the animals remained in their individual home pens (details below). 

### 2.3. Temperament Assessment

The individuals’ temperament traits were assessed during three tests across time and situations: novel object test, novel-environment (open-field test) and anti-predator test, following protocols from refs [39,48]. On test days, food was delivered to individuals only after all pacas had been tested. During the novel object test, the animal was attracted to the back of the home pen (solarium area) (Figure 1A) using a voice command from one research assistant, and then the keeper introduced a plastic ball (0.2 m of diameter) into the home pen from the front door. Both the keeper (male) and research assistant (female) were familiar to the animals. The keeper interacted with the animals daily and the research assistant habituated the animals to her presence in a previous behavioural study so as to minimise any effects of sex [49] and familiarity [50] of humans on experimental results. We chose the ball to perform the novel object test because the animals had never seen this object before, as is expected for this kind of test [2]. Additionally, the ball is not a dangerous object, and it is normally used for environmental enrichment for several mammals [51]. The test started when the paca orientated its head to the ball and appeared to first see it, and lasted for 30 s. This relatively short time window was enough for the judges to rate the animals’ first reactions to a novel object. After the end of the test, the keeper collected the ball, cleaned it with a damp cloth soaked with a solution composed of 70% ethanol and 1% acetic acid [52], and restarted the process in another pen. All pacas were tested on the same morning from 08:00 to 10:00 h, and the test order was randomly determined. The zinc walls of the home pens prevented animals seeing others being tested. Each animals’ reactions were video recorded using the digital camcorder cited above, fixed on a tripod, and placed in front of the pen’s chain-link door. Four months after we conducted the novel object test, all pacas were individually submitted to the novel-environment test on the same day from 08:00 to 10:00 h in the morning. For this test, a square open field test arena on a dirt floor measuring ~10 m^2^ (Figure 1C) was used. Each individual was transported using a wooden cage (0.6 m length × 0.4 m width × 0.3 m height) from its home pen (Figure 1A) to the open-field test arena. The transport cage door was opened, releasing the individual into the open field. The test lasted 30 s, and after the end of this period, the individual was captured and transported back to its home pen (Figure 1A). After an interval of 15 min, the same procedure was carried out with the following paca. During this interval, the keeper sprayed the arena with a solution composed of 70% ethanol, 1% acetic acid to mask the smell of males, as recommended by McGuire et al. [52]. The test order was randomly determined. The same digital camcorder cited above was fixed on a tripod 1.6 m high above one extremity of the arena to continuously video-record the animals’ behaviour during this temperament test.

The anti-predator test was performed two months after the novel-environment test. In this test, each paca was transferred from the home pen (Figure 1A), using the transport cage described previously, to the open field arena (Figure 1C). We opened the open field arena door, releasing the paca inside, and then waited until the animal appeared to be relatively calm exploring the open field arena, without fur bristling, and showing a normal breathing rate, which lasted approximately five minutes. Then the keeper, standing outside the open field arena, introduced into the arena a predator-like stimulus (the net used to capture the pacas for handling and transport). Only when the paca’s head was turned towards the net was it moved towards the animal for 30 s. This procedure would equate to a typical husbandry or predator event that pacas usually respond to by running away, panting, and showing piloerection. The animals had never experienced true predation, although they did experience human predator-like cues [53] when they were caught to be weighed every two months, and for occasional veterinary evaluation in the case of injuries or sickness. The animals always attempted to avoid being caught; therefore, we chose to use the capture net to stimulate the expression of defensive behavioural patterns, such as escape, during the chase and forced contact tests.

Thereafter, the keeper caught the paca with the net and transported it back to its pen. After an interval of 15 min, the same procedures were carried out with the next individual. During the interval, the keeper sprayed the arena with the same solution described above, following McGuire et al.’s recommendations [52]. The same digital camcorder cited above was fixed on a tripod 1.6 m high above one extremity of the arena to continuously video-record the animals’ behaviour during this temperament test. All pacas were tested on the same morning from 08:00 to 10:00 h, and the test order was randomly determined.

Measures of paca temperament were inferred by observing the individuals’ reactions over the three different challenging tests based on the 30 s video footage recorded during each context, following Wemelsfelder et al. [54]. To this end, the behaviour shown by each paca in the 30 s video footage of each temperament test was rated using a subjective rating scale (details below) by three volunteers and animal behaviour experts, not authors of this study. The experts did not participate in data collection and individually watched the 30 s video footage, blinded to the identity of the animals. 

The experts used an analogue scale to rate the animals’ reactions in terms of 14 adjectives. The list of adjectives contained an equal number of adjectives that we inferred to reflect relatively positive and relatively negative states. The seven adjectives that reflected positive states were: “active”, “curious”, “calm”, “docile”, “relaxed”, “bold”, and “satisfied”; while the seven ones that reflected negative states were: “fearful”, “agitated”, “tense”, “anxious”, “apathetic”, “shy”, and “stressed” [54,55]. The adjectives were chosen based on two researchers’ (S.S.C.N. and S.L.G.N.-F.) previous experience of the behaviour of this species, and each adjective was clearly defined (Table 1). For each adjective, the judges were instructed to mark a point on a 125 mm line with a minimum value (0) at the left end of the line representing absence of the behavioural characteristic and, at the right end, the maximum value (125) representing the most intense manifestation. 

### 2.4. Training Sessions for the Judgment Bias Test (JBT)

To train each animal for judgment bias tests (JBTs), we followed the procedures described by ref [57]. In this training session, each paca was transferred, using the transport cage described previously, to the individual cages described above. The training occurred individually, and always occurred in the morning one hour before the usual feeding time to avoid animals’ being satiated and not motivated by the food reward. At the beginning of each trial, the paca was attracted to the back of the cage by the keeper who used a voice and hand command. Thereafter, the individual was trained to move (“go”) from the back to the front of the cage when a positive auditory cue was sounded by a researcher positioned behind the plastic screen mesh; the positive conditioned stimulus was a 3 s whistle (CS+, Freq_average_ 3 kHz; 110.1 dB). Every time the paca showed the correct response, it received a reward provided by another researcher (V.A.) who was also positioned behind a plastic screen mesh. If the paca reached the reward site within 30 s, it received a slice of banana, weighing around 5 g. We considered that the animal had learned this command when showed at least 70% of correct “go” responses after CS+.

After all animals had learnt to “go” in response to the CS+, they were then trained to “no-go”, i.e., to remain at least 1.0 m from the front of the cage (reward site) for 30 s, when a different 3 s auditory cue was sounded; the negative conditioned stimulus was the sound of a caxixi, a percussion instrument (CS−, Freq_average_ 9 kHz; 80.4 dB). If the paca approached the front of the cage within 30 s after the emission of the CS−, the individual received three bouts of short water jets, each bout lasting 2 s with equal 2 s interval between bouts. The burst was targeted to the cage’s front wall, where the reward was delivered only after the CS+. We measured all sound pressures about 1.0 m from the sound sources. 

Training on the judgment bias task involved exposing each paca to 10 CS+ training trials per day for 10 consecutive days, totalling 100 trials, followed by 10 CS− training trials per day for 13 consecutive days, because animals learned the “go” response to the CS+ faster than the “no-go” response to the CS− cue. The training phase was completed when animals achieved a learning criterion of at least 70% correct responses in CS+ and CS− cues. Following that, pacas were exposed to seven further training sessions, one session per day, each containing a mix of 10 CS+ and 10 CS− cues presented in a randomly determined order (a total of 70 CS+ and 70 CS− trials per animal). Throughout all these sessions, pacas showed an average (±standard deviation) of 94.8 (±10.3)% and 82.4 (±16.1)% of correct responses for CS+ and CS−, respectively. 

### 2.5. Judgment Bias Test (JBT)

Following successful training, and 24 h before the MDTB, individuals were submitted to the first JBT. Twenty-four hours after the MDTB, the second JBT was carried out. The JBT always occurred in the morning and, in addition to presentation of the standard CS+ and CS− training stimuli, included a 3 s “ambiguous” auditory cue; the “ambiguous” cue was the sound of a drumstick hitting an aluminium plate (CS_A_, Freq_average_ 6 kHz; 62.8 dB). After CS_A_, “go” or “no-go” decisions were not rewarded or punished. We presented this cue to probe decision-making under ambiguity and to investigate whether animals responded to this cue as if predicting the reward—“optimistic” response—or punishment—“pessimistic” response [35,57]. Each test comprised 10 trials of each of the three cues: CS+, CS−, and CS_A_ per animal. Trial order (CS+, CS− or CS_A_) was randomly determined by the drawing of lots. In each trial, we recorded whether the animal reached the front of the cage (“go” response) within 30 s, or whether it remained at least 1.0 m from the reward location for 30 s (“no-go” response), following Oliveira et al. [57]. Animals were returned to a start location (back of the cage) prior to the next trial by using a voice and hand command. After an interval of 15 min, we carried out the same procedure with the next individual. Test order was randomly determined. We recorded the individuals’ responses by marking “go” or “no-go” on a paper sheet after each cue. 

### 2.6. Modified Defence Test Battery—MDTB

The MDTB was conducted in the morning between 06:30 and 10:30 h. Each animal was submitted to a battery of four consecutive tests: (a) novel-environment; (b) chase; (c) forced contact; and (d) foraging–eating in a novel environment, to assess their defensive behaviour. This methodology, usually called the mouse defence test battery, was modified from refs [31,58], following Nogueira et al. [47]. All pacas were submitted to the four consecutive tests without intervals between them. In this defensive test battery, we used the capture net to simulate the expression of defensive behavioural patterns facing a predator-like cue in two of the tests (chase and forced contact tests, details below). The order in which individuals performed the MDTB was randomly selected.

To start the MDTB, each paca was transferred to the test arena using the described transport cage. The novel-environment test started after the transport cage door was opened and the paca remained free to explore the test arena environment for 20 min. During this test and the following ones, the behaviour of the animals was video recorded. Subsequently, watching the videos, a single observer—an animal behaviour expert but not an author of this study—recorded the amount of time pacas spent on sniffing, abnormal behaviour, roaming, climbing, freezing, running, and raising forelegs (Table 2). Thereafter, the same observer determined the proportion of the time pacas spent on each one of these behaviour patterns.

For the chase test, the procedure took place after the end of this 20 min period, at which point a standard capture net was placed inside the arena by a handler who was outside the arena and hidden by the arena walls. The handler placed the net on the opposite side to where the paca was and, only when its head was turned towards the net did the handler move it at a speed of 0.3 m/s, pursuing the animal. The first chase procedure finished after the net came approximately 0.5 m from the paca, followed by net avoidance and/or a defensive threat/attack towards the net. The chase procedure was repeated two more times. Watching the video-recorded images, a single observer determined the individuals’ flight initiation distance (FID) and flight speed (FS). The FID was calculated from the moment the animal started to react to the capture net. The observer used the chalk lines made on the test arena floor as markers to estimate this distance using the metric system. The FS was also estimated with the chalk lines and a simple chronometer to measure the distance covered by the animal per unit time. All video footage analyses were made by the same observer to avoid measurement bias. Thereafter, we determined the mean FID and FS values from the three chases in each test. Both measures were scored as zero for animals that did not run away from the threat. The observer also determined the time individuals spent on freezing and running behaviours. Forced contact test: immediately following the end of the previous test, both guillotine doors were closed (Figure 1). Thereafter, the handler moved the net towards the paca as in the previous test, but the paca was unable to flee because both guillotine doors were closed. The test ended when the animal expressed one of the following defensive behaviours: defensive threat/attack towards the capture net, or net avoidance followed by flight. As described in the chase test, the individuals’ flight initiation distance and flight speed (details below) were determined alongside the time it spent on freezing and running. For the final “foraging–eating in a novel environment” test the procedure was as follows: immediately following the previous test, we opened the guillotine doors. Thereafter, the keeper put 300 g of banana (*Musa* spp.) on the test arena floor at the opposite side to where the animal was and left the animal free to walk and explore the environment for 20 min. Banana was chosen because it is a highly favoured food of the paca [23]. During this test, the difference between the weight of banana offered and its weight at the end of the session was used to calculate food consumption. During this test, it was also determined the amount of time pacas spent on the described behaviours (Table 2). 

### 2.7. Endocrine Stress Responses: Faecal Glucocorticoid Metabolites

A first set of faecal samples was collected from each individual before the first temperament test (novel object test—ball) and the MDTB. To determine the average faecal glucocorticoid metabolite (FGCM) concentration in samples collected before the MDTB (basal level), the keeper cleaned the cages at 06:00 h each morning and at 10:30 h each day, and collected fresh faeces excreted between 06:00 and 10:30 h during three consecutive days. The second set of faecal samples was collected 24 h after the end of the MDTB. The 24 h interval was chosen because peak excretion of glucocorticoid metabolites in paca faeces appears to occur around this time after acute stress [59]. Faecal samples were stored in identified plastic containers and refrigerated at −20 °C for further analysis [60]. Prior to analysis, the three consecutive days of pre-MDTB faecal samples from each individual were thawed, homogenized. and pooled, while the second set of faecal samples (post-MDTB) were just thawed and homogenized. Then, 1–2 g from each individual in each situation (before and after the MDTB) was sub-sampled and stored at −20 °C in preparation for freeze-drying (FreeZone^®®^ Plus 4.5 Liter Cascade Benchtop, LABCONCO), following Wasser et al. [61]. 

The extraction and measurement of FGCM in these freeze-dried samples were carried out at the Núcleo de Pesquisa e Conservação de Cervídeos (NUPECCE) at Faculdade de Ciências Agrárias e Veterinárias da Universidade Estadual Paulista (FCAV—UNESP Jaboticabal, Brazil), according to the methodology described by Graham [62]. Steroids were extracted by adding 2.0 mL of 90% methanol (10% water) to ~0.5 g of dried faeces, for the best recovery of glucocorticoid metabolites, as determined by Coradello et al. [60]. Thereafter, the tubes were vortexed for 30 s, followed by shaking for 12 h on a horizontal shaker (Mod. AP22^®^—Phoenix Ltd.a—Araraquara, Brazil), and vortexed again for 10 s. Then, the tubes were centrifuged at 1500 rpm for 20 min. The supernatant (containing hormones and metabolites) was placed in identified plastic tubes and stored at −20 °C until assay. We adjusted the volume of the methanol to the same proportion as the weight of the sample (0.25 g of sample in 2.5 mL) of methanol for samples that did not have sufficient material. This adjustment was performed in just 11% of the analysed samples (*n* = 178).

The concentration of FGCM was measured using enzyme immunoassays (EIA), with the antibody cortisol–HRP conjugate was obtained from Ms. Coralie Munro (California University, Davis, CA, USA). Cross-reactivity for the antibody was 100% with cortisol, 9.9% with prednisolone, 6.3% with prednisone, and 5.0% with cortisone [63]. Intra- and inter-assay coefficients of variation were 7.0 ± 5.3% and 9.2 ± 3.8%, respectively, validating the assay’s precision [64,65]. 

The immuno-enzymatic assay was carried out on NUNC ELISA plates (Thermo Scientific, Waltham, MA, USA), which were coated with 50 mL of antibody diluted in a coating buffer (0.05 M NaHCO3, pH 9.6) and stored for about 15 h at 4 °C. Unbound antibodies were rinsed away with a wash solution (0.15 M NaCl, 0.05%). Then, 50 µL of the dilutions of each sample, dilutions of the standard curve, and the controls (high and low; diluted with EIA buffer) were added to each well. After that, the corresponding HRP conjugate was added for the corticosterone or cortisol assays and incubated for two hours (for corticosterone) or one hour (for cortisol) at 24 °C. Subsequently, the plates were washed, 100 mL of the ABTS solution was added to each well, and the plates were placed on a shaker until the white wells arrived at an optical density of 0.7. Absorbance was measured under 405 nm light with the plate reader (Multiskan Ascent—Thermo Scientific). FGCM concentrations were expressed as ng per g of dry faeces. 

### 2.8. Data Analyses and Statistics

To analyse the temperament ratings, we followed the methods described by Feaver et al. [55], which are suitable for a small number of individuals, as was the case in our study. First, the ratings of the three judges were converted to z-scores ((individual score—mean)/SD) to reduce the influence of distributional effects. We then determined the inter-observer agreement using Kendall’s coefficient of concordance (*W*), for each adjective and test independently. Further analysis only involved those adjectives that showed inter-observer coefficients of concordance greater than or equal to 0.70, following the procedures described by Feaver et al. [55]. For each of these items, we calculated the mean value of the observers’ ratings for each paca in each one of the three challenge tests (novel object, novel-environment test, and predator-like tests). Thereafter, Kendall’s coefficients of concordance (*W*) of each adjective among the three tests were determined. We selected for further analysis only the adjectives that showed inter-test coefficients of concordance higher than 0.70 (*W* > 0.70). The selected adjective mean z-scores across the three tests were then tested for correlations by Spearman’s rank correlation tests. Following that, we determined the mean z-score of the highly correlated adjective descriptors (r_S_ > 0.70 and r_S_ < −0.70) as indicators of specific temperament characteristics in *Cuniculus paca* that showed cross-time and cross-situation consistency.

All video-recorded images from the MDTB and JBT were coded by a single observer, who was blind to the experimental treatments during video-analysis, using the software CowLog 3.0.2 [66]. Before the analysis, we evaluated whether all data fulfilled the parametric requirements of normality of residuals and homogeneity of variance. The data did not fulfil assumptions even after transformation, and therefore Spearman rank correlations were used to test the association between temperament metrics and the amounts of time each paca spent on: roaming and raising forelegs during novel-environment and foraging–eating tests; abnormal behaviour during foraging–eating tests; and freezing, flight speed (FS), and flight initiation distance (FID) in the chase and forced contact tests of the MDTB. We thus aimed to investigate whether and how our temperament metric was associated with exploration (roaming and raising forelegs) in novel circumstances [67,68], vulnerability to displaying abnormal behaviour [69], and boldness (FID) in response to challenge, a characteristic that may explain differences in where individuals settle in relation to humans [70,71].

To determine if animals had learnt the discrimination task during the training days prior to the judgment bias test, we ran binomial tests on individual data (probability of correct response on each trial = 0.5), following Oliveira et al. [57]. For the test sessions, we determined the number of trials in which each paca reacted with the “go” response within 30 s for each cue type. The cognitive bias data, which were not normally distributed, were log-transformed before being analysed using a general linear model (GLM) with repeated measures followed by post-hoc Tukey tests. The phase the judgement test was applied (before and after the MDTB), cue type (CS+, CS−, CS_A_) were the within-subject factors, and we examined their effects, including interactions, on the number of “go” responses made. A judgement bias index was calculated for each test separately as the number of negative responses to ambiguous cues subtracted from the number of positive responses, resulting in values ranging between −1 and 1, where values above 0 indicated an overall positive judgment and “optimistic” interpretation of the ambiguous cue [72]. To test behavioural plasticity, we compared the same behaviours (Table 2) recorded during similar tests: novel-environment vs. foraging–eating test; and chase test vs. forced contact test by using the Wilcoxon matched pairs test. Friedman ANOVA was used to compare the concentrations of faecal glucocorticoid metabolites (FGCM) in the faecal sample collected before and after the MDTB. We used Spearman’s rank correlation to test the association between temperament dimension with the judgement bias indices and the FGCM concentrations before and after the MDTB, and with the increase in the FGCM concentration (difference between FGCM concentrations in samples collected after the MDTB compared to the individuals’ pre-MDTB levels). FGCM concentrations, which were not normally distributed, were log-transformed before the last analysis. The software Statistica 7.0 (StatSoft, Inc 1984–2004) was used for all analyses, and *p*-values less than 0.05 were considered statistically significant for all analyses. 

## 3. Results

### 3.1. Temperament Tests

There was concordance amongst the raters for 11 of the 14 adjectives (*W* > 0.70; Table 3). Of these, concordance was verified (*W* > 0.70) across the three challenge tests for eight adjectives: “active”, “agitated”, “calm”, “curious”, “fearful”, “tense”, “bold”, and “anxious” (Table 3). 

Spearman’s rank correlations amongst these eight adjectives resulted in a single group of highly correlated descriptors (r_s_ > 0.70; Table 4), which were combined to yield a temperament dimension named “restless” in the following way:

“Restless”: obtained from the mean z-scores of the adjectives (fearful + agitated + tense)/3.

Pacas 5, 7 and 8 were considered as “restless”. In turn, pacas 1, 2, 3, 4, and 6 were judged as “non-restless” (Figure 3).

### 3.2. Modified Defence Test Battery (MDTB) Behaviour and Its Relationship to Temperament 

During the novel-environment test, there was a negative correlation between the individuals’ mean z-scores on the “restless” temperament dimension and the time they spent on exploratory patterns (roaming + raising forelegs, r_Spearman_ = −0.89, *p* = 0.03) (Table 5). There was also a positive correlation between the individuals’ “restless” scores and the time they spent on abnormal behaviour (r_Spearman_ = 0.71, *p* = 0.04) (Table 5). There was, however, no correlation between the individuals’ “restless” scores and the time they spent on trying to escape (running + climbing, r_Spearman_ = 0.24, *p* = 0.57) (Table 5).

During both chase and forced contact tests there were no correlations between “restless” scores and the flight initiation distance (FID) (chase test: r_Spearman_ = −0.35, *p* = 0.40; forced contact test: r_Spearman_ = −0.19, *p* = 0.69); flight speed (FS) (chase test: r_Spearman_ = 0.29, *p* = 0.49; forced contact test: r_Spearman_ = −0.17, *p* = 0.87); and freezing (chase test: r_Spearman_ = 0.62, *p* = 0.10; forced contact test: r_Spearman_ = 0.48, *p* = 0.23) (Table 5). Finally, during the foraging–eating test, all pacas ate the total amount (300 g) of banana available. The pacas took from 26 s to 8 min to start eating the banana and 5 to 11 min to finish eating (Table 5). There were no correlations between the individuals’ “restless” scores and the time pacas spent on exploratory behaviour (roaming + raising forelegs, r_Spearman_ = 0.33, *p* = 0.42) and the time they spent on trying to escape (running + climbing, r_Spearman_ = 0.24, *p* = 0.57) (Table 5). During this test, pacas did not perform abnormal behaviour. 

### 3.3. Evaluation of Behavioural Plasticity

Comparing the same behaviour patterns performed during novel-environment and foraging–eating tests, there was a decrease in the total time spent on the exploratory patterns (Wilcoxon matched pairs test Z = 2.52, *n* = 8, *p* = 0.01, Figure 4A) and in climbing (Wilcoxon matched pairs test Z = 2.52, *n* = 8, *p* = 0.01, Figure 4B). Furthermore, comparing data from the chase test to the forced contact test, both the flight initiation distance (FID) and the flight speed (FS) increased in the latter test (FID: Wilcoxon matched pairs test Z = 2.17, *n* = 8, *p* = 0.02, Figure 5A; FS: Wilcoxon matched pairs test Z = 2.20, *n* = 8, *p* = 0.03, Figure 5B). In contrast, there were no differences between the amounts of time spent on the patterns of freezing (Wilcoxon matched pairs test Z = 0.70, *n* = 8, *p* = 0.48) and running (Wilcoxon matched pairs test Z = 0.98, *n* = 8, *p* = 0.33) between the chase and forced contact tests (Table 5). 

### 3.4. Judgment Bias Test (JBT) and Temperament 

The number of correct responses that pacas made to the CS+ cue (“go”) and responses that they made to the CS− cue (“no-go”) increased from the first to the last of the final days of training, showing a learning process. From the second day of training, all individuals showed more than 80% of correct “go” responses to the CS+ cue (Figure 6A), while they took up to 12 days to show more than 70% of correct “no-go” responses to the CS− cue (Figure 6B), indicating that they had learnt how to discriminate the task. There was, however, no correlation between “restless” scores with the number of days to reach at least 70% correct answers to CS− (r_P_ = −0.11, *p* = 0.80).

Although the statistical model revealed a significant interaction between cue type and phase (F _2,14_ = 4.43, *p* = 0.03), the post-hoc tests showed that, in both phases (before and after the MDTB), the mean log transformed proportions of “go” responses to the CS− cue (0.7 ± 0.9) were lower (Ps < 0.0002) than the mean proportions of “go” responses to the CS+ (1.7 ± 0.7) and CS_A_ (1.7 ± 0.7) cues, which occurred in similar proportions (*p* = 0.99, Figure 7). Before the MDTB, there was no correlation between the judgement bias index with “restless” scores (r_Spearman_ = 0.08, *p* = 0.85). However, after the MDTB, there was a near-positive significant correlation between cognitive bias indexes of the judgment bias test with “restless” scores (r_Spearman_ = 0.70, *p* = 0.05).

### 3.5. Concentration of Faecal Glucocorticoid Metabolites and Subjective Dimensions of Temperament 

The average FGCM concentration in samples collected before the MDTB (basal level) was lower than the concentrations in samples collected after the MDTB (64.3 ± 69.6 vs. 333.5 ± 601.3 ng/g of dry faeces, respectively (Wilcoxon matched pairs test Z = 2.52, *n* = 8, *p* = 0.01). Both FGCM concentrations in samples collected before and after the MDTB were not correlated with the “restless” temperament scores (r_Spearman_ = 0.52, *p* = 0.18 and r_Spearman_ = 0.40, *p* = 0.32, respectively). There was also no correlation between the “restless” scores and the increase in FGCM concentration over time (r_Spearman_ = 0.43, *p* = 0.34) (Table 5).

## 4. Discussion

Observers rated paca as showing individual consistency in the correlated measures of “fearfulness”, “tension” and “agitation” across three different test contexts and over 180 days. The resulting “restless” temperament score was correlated with some of the defensive responses displayed throughout the modified defence test battery (MDTB), which elevations in faecal glucocorticoid metabolites indicated was indeed stressful, and with a measure of judgement bias following the MDTB. Moreover, as expected, pacas considered the ball an attractive object and the net as a real threat which they avoid. The obtained results support these statements, because all pacas interacted with the ball in the novel object test and all of them fled from the net in both chase and forced contact tests. 

During the novel-environment phase of the MDTB, pacas scoring low on the “restless” temperament dimension spent more time roaming and raising forelegs, which are exploratory behaviour patterns also observed in other rodent species when in unfamiliar surroundings (e.g., *Rattus norvegicus* and *Georychus capensis* [73]). In contrast, individuals judged as more “restless” spent more time eating grass, which we considered to be an abnormal behaviour for the species. Free-ranging pacas mainly eat fruits and small amounts of tree leaves, while there is no record of grass in the species’ natural diet [74]. Blanchard [75] states that unexpected feeding disturbance during defence test trials is usually interpreted as a failure in risk assessment in rats. Therefore, the abnormal behaviour recorded here by pacas judged as “restless” may be interpreted as a sign of fear-like or anxiety-like states [69,75,76,77].

Overall, these findings indicate a link between paca “restless” temperament scores and behaviour in the novel-environment phase of the MDTB. One potential reason for this is that this phase of the MDTB bore some resemblance to the novel object and novel-environment parts of the temperament tests, and hence elicited the same type of behavioural responses. However, the temperament tests were extremely short (30 s each) compared to this phase of the MDTB test (20 min), and therefore it is also possible that they were able to detect a core paca temperament characteristic that predicted defensive responses. 

During the remaining chase, forced contact, and foraging–eating phases of the MTDB, “restless” temperament scores did not correlate with recorded paca behaviour. This may be because paca perceived the short temperament tests differently from the longer MDTB tests and adjusted their responses accordingly and flexibly, such that behaviour in one test type was not a predictor of that in the other. On the other hand, it is possible that a larger sample size would be required to detect correlations.

During the training for the judgement bias test, all individuals learnt both commands (“go”/”no-go”) after the 12th day. However, paca learnt the positive conditioned stimulus (CS+) quicker than the negative conditioned stimulus (CS−). In both judgement bias tests performed before and after the MDTB, all paca treated the “ambiguous” stimulus (CS_A_) as more similar to the CS+ than the CS−. In addition, there were no differences in the cognitive bias indexes, i.e., the proportion of “go” responses to the CS_A_ cue, determined in both judgement bias tests. One possible explanation for this is that the sharp onset and high frequencies of the drumstick/aluminium plate CS_A_ sound was perceived by paca as being more similar to the CS+ whistle sound than the “shaking” sounds produced by the CS− caxixi. This is one potential drawback of using qualitatively different sound cues in judgement bias testing. Here, they were used to enhance speed of training for this wild species under outdoor conditions where the potential for interfering sounds was high (cf. previous studies of peccary: refs [57,78], and see also ref [79]). 

An alternative explanation is that overhunting of paca has generated a generally threatening environment for this species [11,24], and that following “state dependent detection theory” (SDDT) [80], this has resulted in paca being more likely to make “optimistic-like” decisions under ambiguity. Contrary to traditional signal detection theory predictions that negative events (e.g., predator presence) should increase the likelihood of avoidance responses, SDDT predicts that they should actually lead to lower levels of caution [81]. SDDT posits that animals exposed to repeated danger in the environment cannot afford to constantly avoid danger because this will inhibit activities such as foraging and hence ultimately reduce energy reserves, fitness, and survival chances [81,82]. Therefore, they are expected instead to become bolder. Such an effect may also explain why hunters easily lure pacas into traps using locally available fruits [25].

Individual “restless” scores were correlated with the judgment bias index, but only in the second judgment bias test which followed the MDTB. Individuals with higher “restless” scores were more likely to make the optimistic-like “go” response to the ambiguous signal (CS_A_). Positive responding to ambiguous stimuli in the judgement bias task is predicted to be more likely in individuals in a more positive affective state [35,36,83]. The current finding appears to contradict this prediction [84]. One potential explanation is that, for example, a mild negative affective state induced by experience of negative events does not just increase an animal’s expectation of subsequent negative events, but also increases their valuation of positive outcomes, and that these conflicting influences combine to determine decisions. Computational modelling of decision data can be used to identify and disentangle these effects [85]. 

Plasticity in paca behaviour under challenge can be inferred from changes in paca responses during different phases of the MDTB. Paca decreased both the time spent on exploratory patterns (roaming and raising forelegs) and escape attempts (climbing pattern) during the foraging–eating phase relative to the earlier novel-environment phase of the MDTB, and ate all the banana delivered in the latter phase, even though this followed relatively intense challenges of being chased by, and then forced into direct contact with, humans. Additionally, pacas increased their flight speed and flight distance across the forced contact test, indicating adaptive adjustment of responses according to differences in perceived risk, as suggested by López and Martin for other species [1]. Comparable results were reported for the Columbian black-tailed deer (*Odocoileus hemionus columbianus*), which showed differences in the flight initiation distance according to predator approach style [82].

The fact that the species is able to rapidly change its behaviour from a very distressed condition to a more positive condition, even feeding naturally after chasing events, is an intriguing result. This behavioural plasticity, however, can act favouring resilience, but also against it. The pacas could face high levels of hunting deaths and could be easily caught. This may occur in situations of high hunting pressure, reported by several authors [12,13,14,15,16], or when other targeted species are reduced in numbers and pacas may then become selected prey. In turn, behavioural plasticity may favour the species’ survival: by not avoiding foraging, even in supposedly dangerous situations, pacas can increase their energy intake, therefore having better fitness, and reproducing more efficiently. In relatively low hunting areas, this may occur, and allow paca to disperse and occupy available empty space in hunted areas through a small-scale source–sink process [86]. This behaviour may contribute to the resilience of this species despite the threat of hunting that it faces. It is also important to consider that such rapid changes in behaviour from a very distressed state to a more positive state might be more common in captive individuals and relatively small individuals born and raised in captivity than under natural conditions. However, it is very difficult, maybe even impossible, to study the plasticity of defensive behaviour of free-ranging pacas. Thus, it would be interesting to carry out further studies on larger samples of captive pacas in order to confirm the results described here. 

Overall, these findings indicate fairly rapid habituation to the testing situation, such that by the time of the foraging–eating phase, there was no evidence of stress-induced suppression of feeding behaviour in the presence of a rewarding stimulus—banana [23]. Again, this has some parallels with SDDT which, as discussed above, predicts that exposure to danger should result in animals becoming less risk-sensitive [80,81]. Concordant results were reported for cichlids (*Archocentrus nigrofasciatus*) [87], which, despite being exposed to a high-risk environment, usually display a low-intensity response to alarm cues. However, this theoretical perspective applies particularly to scenarios in which the animal has experienced longer-term danger or has clear predictions that the future is likely to remain dangerous, and this may be less applicable to the short-term tests considered here.

During both chase and forced contact phases of the MDTB, pacas showed freezing and running responses, typical rodent escape behaviours [88], and these did not differ between the two phases. Emergency responses facing a predator are likely to be strongly selected and underpinned by innate defensive survival circuits, as has been revealed in rats [89], and hence little individual variation is expected within a species. Spotlighting, baiting, and waiting in a tree for individuals to come at night are common methods used to hunt pacas [12]. In this kind of hunting, freezing is described by hunters as related to an increase in their chances of killing pacas, because “frozen” individuals become easier to shoot in contrast to pacas that flee immediately after seeing the flashlights. Hunters also commented that the freezing behaviour varies according to the type of flashlight used during hunting. Therefore, it would be very interesting to study pacas’ behaviour at night to assess how they differ after facing predator-like tests using flashlights with different light intensities.

Paca showed an increase in the concentration of faecal glucocorticoid metabolites (FGCM) from samples collected before to those collected after the MDTB. However, in contrast to what we expected, there were no correlations between FGCM concentrations and the paca’s temperament traits. Relationships between endocrine response and temperament traits have been shown in several species (beef cattle [90]; humans [91]; pigs [92]; and common marmosets [93]). There are, however, some other studies in which the lack of correlation between temperament traits and endocrine responses was also verified (rats [94]; greenfinches (*Chloris chloris*) [95]; and wild marmots (*Marmota marmota*) [96]). Therefore, the increase in the concentration of faecal glucocorticoid metabolites in pacas after they faced highly stressful situations is independent of temperament phenotypes. However, for better interpretation, further studies need to investigate the relationship between the hypothalamic–pituitary–adrenal axis and the freezing–fight–flight responses in pacas. 

Despite attempts to design a robust experiment within the constraints of our lab environment, our study inevitably has limitations that prevent us from drawing firm conclusions about the paca’s resilience to over-hunting by humans. These include the small sample size, the use of qualitatively different sound cues in the judgement bias tests that may have resulted in all animals treating the “ambiguous” cue as being more similar to the positive one, and the fact that the predictive reliability of the temperament tests used here still requires further validation [97,98]. Thus, further studies must be conducted to improve and confirm our conclusions. 

## 5. Conclusions

The pacas studied herein showed cross-time and context stability in a temperament dimension labelled “restless”. Individual “restless” scores predicted responses to novelty, although not to simulated chasing and capture by humans in a separate modified defence test battery—MDTB. More “restless” animals showed a greater proportion of positive responses (to “go”) to an ambiguous cue during the JBT after the MDTB. Plasticity in defensive behaviour was inferred from changes in behavioural responses and apparently rapid adaptation to challenge in the different phases of the MDTB. The results indicate that both temperament and behavioural plasticity may play a role in influencing paca responses to risky situations. Therefore, our results allow us to suggest that individual differences and the consistency of behavioural responses displayed by paca toward the MDTB challenges, together with the species’ ability to modulate their responses according to risk levels and individual variability in learning the negative conditioned stimulus (CS−), may reflect a generally flexible and successful defensive behavioural response that underpins the paca’s survival despite the threat of overhunting throughout its range.

## Figures and Tables

**Figure 1 animals-11-00293-f001:**
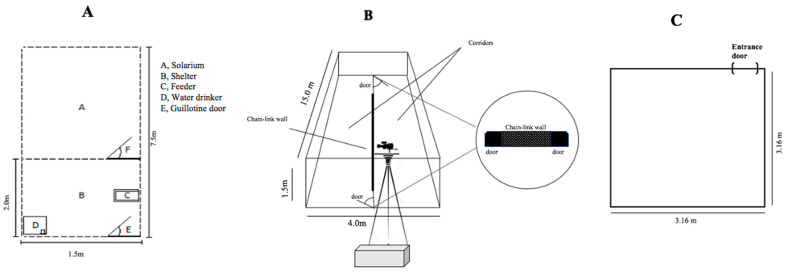
Diagrams of the pacas’ home pen (**A**); arena used for the modified defence test battery (MDTB) (**B**); and open field used during temperament assessment (novel-environment test) (**C**).

**Figure 2 animals-11-00293-f002:**
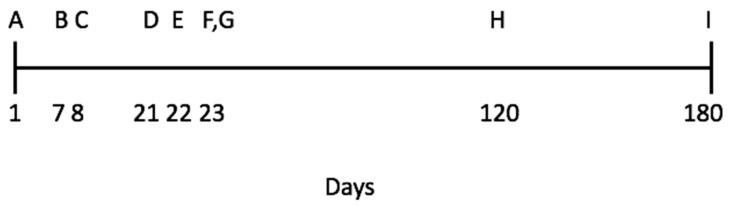
The experiment timeline. This study was conducted over 180 days following this timeline sequence: A: collection of faecal samples to determine basal faecal glucocorticoid metabolite concentration; B: temperament assessment (novel object test); C: training sessions for the judgement bias test (JBT); D: the first JBT; E: modified defence test battery (MDTB); F: the second JBT; G: the second collection of faecal samples to determine faecal glucocorticoid metabolite concentration, (F and G both occurred 24 h after the MDTB); H: temperament assessment (novel-environment test); I: temperament assessment (anti-predator test).

**Figure 3 animals-11-00293-f003:**
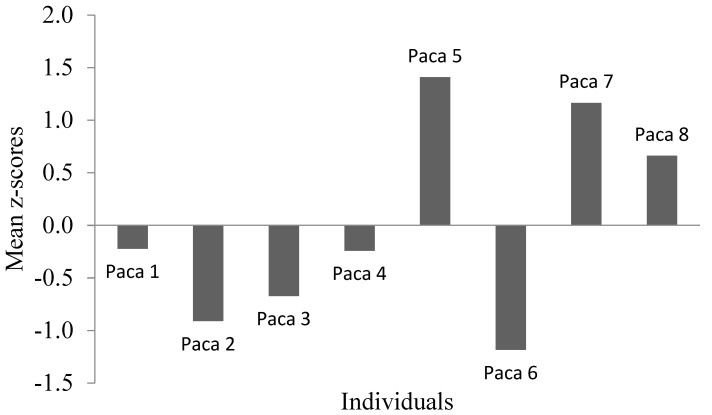
Individuals’ mean z-scores in the “restless” temperament dimension of paca males (*n* = 8).

**Figure 4 animals-11-00293-f004:**
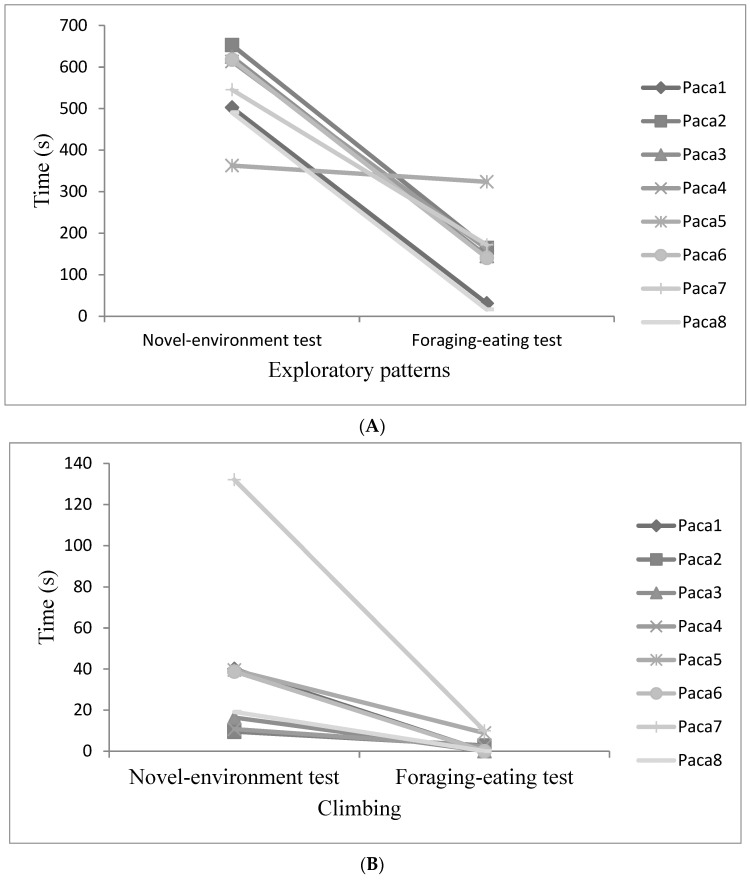
Time pacas (*n* = 8) spent on the exploratory behavioural patterns (roaming, sniffing, and raising forelegs) (**A**) and climbing pattern (**B**) during novel-environment and foraging–eating tests.

**Figure 5 animals-11-00293-f005:**
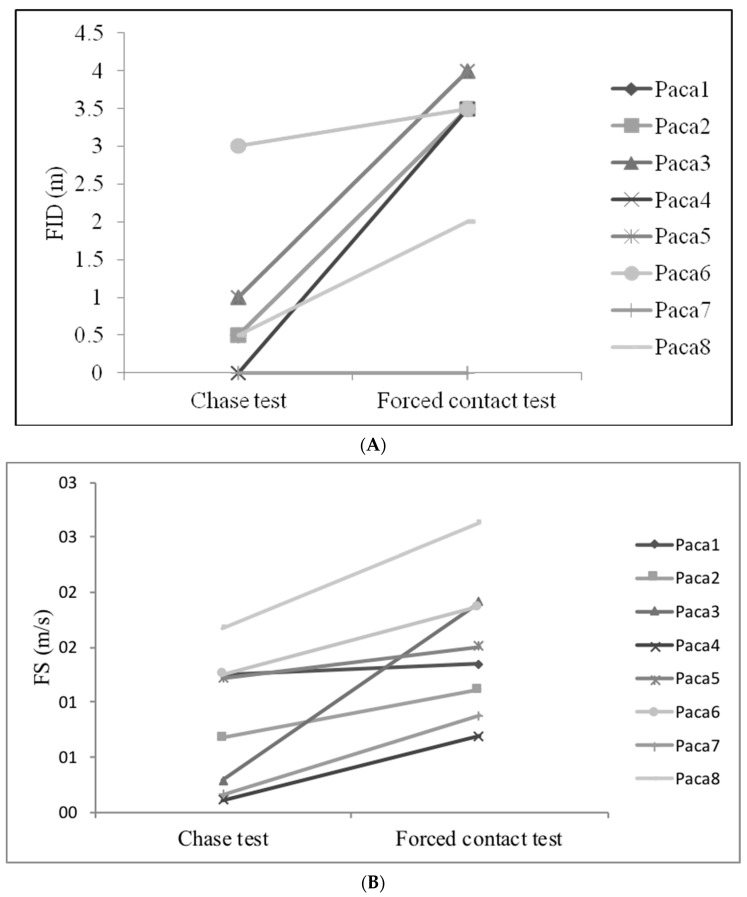
Flight initiation distance (FID) (**A**) and flight speed (FS) (**B**) of pacas (*n* = 8) during the chase and forced contact tests.

**Figure 6 animals-11-00293-f006:**
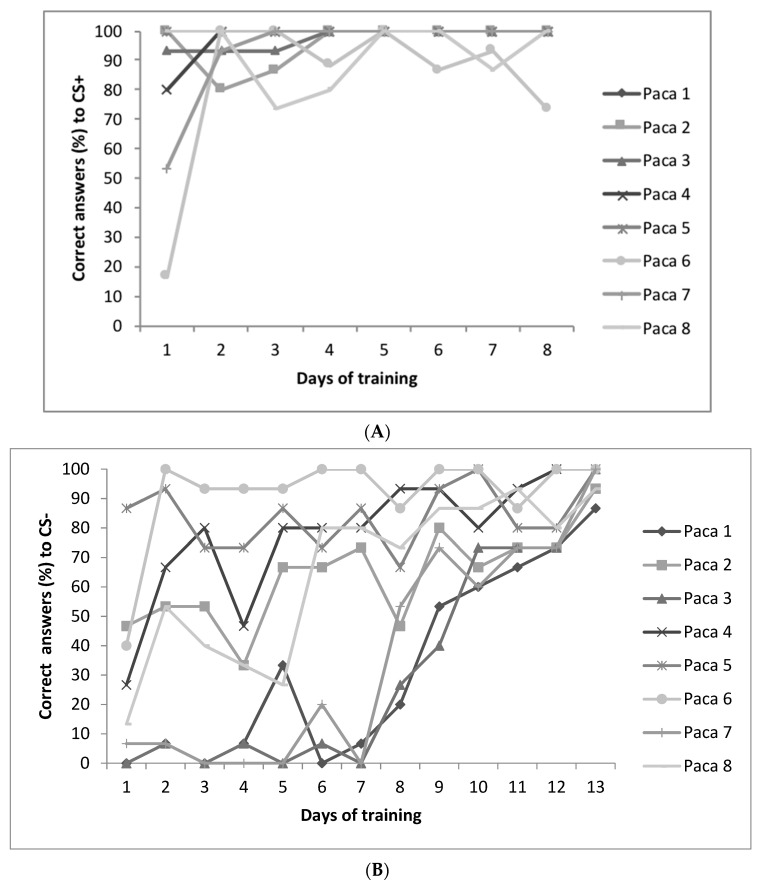
Number of correct responses made by pacas (*n* = 8) during training sessions to CS+ cue (**A**) and to CS- cue (**B**).

**Figure 7 animals-11-00293-f007:**
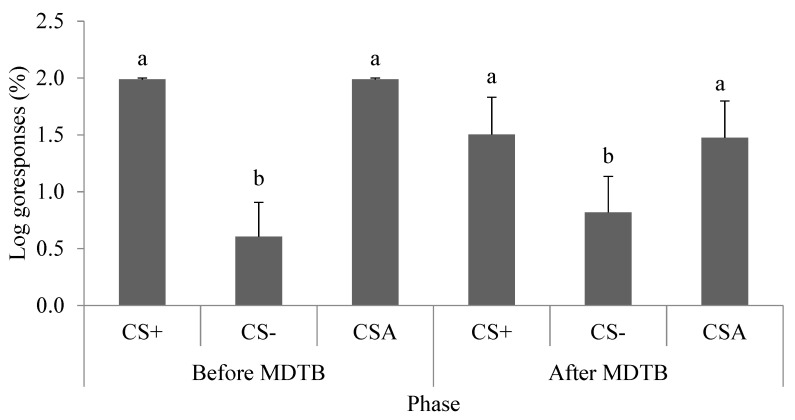
“Go” responses (%) after cues throughout the judgment bias tests performed before and after the modified defence test battery (MDTB). CS+, positive auditory cue; CS-, negative auditory cue; CS_A_, “ambiguous” auditory cue. Bars with different superscripts are significantly different according to Tukey post-hoc tests.

**Table 1 animals-11-00293-t001:** List of adjectives definitions used to evaluate the paca’s temperament traits.

Adjective	Definition *
Active	The active individual moves around a lot or engages in a range of different activities, such as walking and foraging
Curious	The individual appears to be investigating its surroundings, smelling the soil or the air
Apathetic	The individual appears to show a lack of interest in or concern about the environment
Fearful	The individual appears to be alert and apprehensive when facing challenging situations, walking slowly, but stopping from time to time
Agitated	The individual appears to be extremely disturbed or excited, moving around rapidly, stopping and walking suddenly
Calm	The individual appears to be peaceful and tranquil, showing relaxed jaw muscles and walking at a regular speed
Tense	The individual shows rigid jaw muscles and bristly hair, as if about to react strongly
Docile	The individual shows cooperative and submissive behavior. The animal shows no bristly hair, no aggression, and relaxed jaw muscles
Anxious	The individual appears to be vigilant and is highly responsive to changes in the environment. It may run or try to hide close to a wall, and move back and forth quickly
Relaxed	The individual appears to be at ease, with relaxed muscles and no bristly hair
Shy	The individual is cautious and tends to hold back in challenging situations
Bold	The individual readily explores its surroundings, even when facing dangerous and challenging situations
Satisfied	The individual appears to be relaxed and calm following an event. It shows relaxed muscles, no bristly hair, and moves at a regular speed
Stressed	The individual is restless, vigilant and shows panting, bristly hair and movement of jaw muscles.

* Definitions were adapted from the Merriam-Webster dictionary [56].

**Table 2 animals-11-00293-t002:** Description of pacas’ behavioural responses during the defence test battery.

Behaviour	Description
Exploratory patterns	
Roaming	The animals move about or travel aimlessly or unsystematically, especially over the arena
Raising forelegs	The paca lifts both forepaws, sniffing the arena walls. The animal seems relaxed, without fur bristled
Trying to escape patterns	
Running	The animal makes fast movement to escape
Climbing	The animal climbs the test arena walls with forepaws, while the fur is bristled
Freezing	Stationary state lasting more than 3 s regardless of posture
Eating patterns	
Eating banana	The paca takes pieces of banana with its mouth, chews, and swallows it
Abnormal behaviour	The paca eats inappropriate food (grass) found on the dirt floor of the arena

**Table 3 animals-11-00293-t003:** Inter-observer and inter-challenge tests (novel object, novel-environment, and anti-predator) using Kendall’s concordance coefficients (*W*) of individual behavioural distinctiveness traits.

	Inter-Observer		Inter-Challenge Tests	
Adjective *	*W*	Significance (*p*)	*W*	Significance (*p*)
**Active**	**0.70**	***p*** **< 0.05**	**0.86**	***p*** **< 0.05**
Relaxed	0.52	n.s.	-	-
**Agitated**	**0.95**	***p*** **< 0.05**	**0.93**	***p*** **< 0.05**
**Fearful**	**0.70**	***p*** **< 0.05**	**0.80**	***p*** **< 0.05**
**Calm**	**0.70**	***p*** **< 0.05**	0.57	*p* < 0.05
**Tense**	**0.72**	***p*** **< 0.05**	**0.98**	***p*** **< 0.05**
**Anxious**	**0.70**	***p*** **< 0.05**	0.64	*p* < 0.05
Satisfied	0.03	n.s.	-	-
Stressed	**0.73**	***p*** **< 0.05**	0.14	n.s.
Nervous	0.60	n.s.	-	-
Apathetic	**0.98**	***p*** **< 0.05**	0.33	n.s.
**Curious**	**0.86**	***p*** **< 0.05**	**0.98**	***p*** **< 0.05**
Shy	**0.90**	***p*** **< 0.05**	0.09	n.s.
**Bold**	**0.99**	***p*** **< 0.05**	**0.99**	***p*** **< 0.05**

* Items in bold type are those in which the inter-challenge tests’ concordance coefficients (*W*) were greater than or equal to 0.70, and thereby qualified for use in further analysis.

**Table 4 animals-11-00293-t004:** Spearman’s coefficient of correlations (r_Spearman_ values) * among the mean ratings of selected adjectives.

	Active	Fearful	Agitated	Tense	Bold	Curious
Active	-	−0.33 (*p* > 0.05)	−0.24 (*p* > 0.05)	−0.19 (*p* > 0.05)	−0.24 (*p* > 0.05)	−0.29 (*p* > 0.05)
Fearful		-	**0.95 (*p* < 0.05)**	**0.88 (*p* < 0.05)**	−0.10 (*p* > 0.05)	0.24 (*p* > 0.05)
Agitated			-	**0.98 (*p* < 0.05)**	0.02 (*p* > 0.05)	0.24 (*p* > 0.05)
Tense				-	0.07 (*p* > 0.05)	0.17 (*p* > 0.05)
Bold					-	0.60 (*p* < 0.05)
Curious						-

* Bold r_Spearman_ values represent rs > 0.70 with significance *p* < 0.05 that were used to combine the “restless” behavioural trait dimension.

**Table 5 animals-11-00293-t005:** The relationship between “restless” temperament scores and modified defence test battery (MDTB) behaviours and faecal glucocorticoid metabolites.

Animal				New-Environment Test			Foraging-Eating Test				Chase Test			Forced Contact Test		
	z-Score ^1^	FGC ^2a^	FGC ^2b^	Exp ^3^	Esc ^4^	Ab. Behav. ^5^	Exp ^3^	Esc ^4^	Ab. Behav. ^5^	Eating Banana	FID ^6^	FS ^7^	Freez. ^8^	FID ^6^	FS ^7^	Freez. ^8^
Paca 1	−0.22	86.0	240.4	502	41	29	31	0	0	639	0.5	1.3	7.6	3.5	1.3	12.0
Paca 2	−0.91	22.6	116.6	659	37	18	165	3	0	498	0.5	0.7	1.2	3.5	1.1	3.6
Paca 3	−0.67	13.7	32.1	632	101	58	145	0	0	362	1.0	0.3	0.2	4.0	1.9	4.4
Paca 4	−0.24	54.6	161.2	616	18	14	162	2	0	324	0.0	0.1	6.1	3.5	0.7	6.0
Paca 5	1.41	34.7	102.9	365	88	102	324	9	0	489	1.0	1.5	29.7	4	1.2	6.6
Paca 6	−1.19	30.4	49.2	632	44	6	141	0	0	365	3.0	1.3	6.9	3.5	1.9	4.4
Paca 7	1.16	44.7	152.7	546	132	56	172	10	0	484	0.0	0.2	31.2	0.0	0.9	19.0
Paca 8	0.66	227.4	1812.8	490	35	52	16	0	0	426	0.5	1.7	4.7	2.0	2.6	1.9

^1^ Mean z-score of the “restless” temperament dimension; ^2^ FGC: faecal glucocorticoid metabolites (ng/g of dry faeces) before (a) and after (b) MDTB; ^3^ Exp: time spent (s) on exploratory behaviours (roaming + raising forelegs); ^4^ Esc: time spent (s) on trying to escape (running + climbing); ^5^ Ab. behav.: abnormal behaviour (s); ^6^ FID: flight initiation distance (m); ^7^ FS: flight speed (m/s); ^8^ Freez.: freezing (s).

## Data Availability

The datasets used and/or analysed during the current study are available from the corresponding author on reasonable request.

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
