# Peer review of "Temperament, Plasticity, and Emotions in Defensive Behaviour of Paca (Mammalia, Hystricognatha)"

_animals, 2021, doi:10.3390/ani11020293_

Round 1

Reviewer 1 Report

Temperament, plasticity and emotions in defensive behavior of paca

Overall this paper is well written with lengthy introduction and descriptions of all the methods used. That said, the methods and their timing are complex and I must say a bit difficult to follow (with a few acronyms to remember for people not used to them).

Specific comments include:

Simple summary: this summary is for lay persons so terms such as individuals’ plasticity and individuals’ affective states may not be clear.

Abstract: Please put abbreviations in brackets after full term (e.g. JBT and MDTB).

Also, as in other places use present tense when talking about what the results ‘indicate’ (L28)

“cross time” probably should be “across time”

L32 Please delete “(to go)” as it just complicates the meaning.

Introduction: L60 an unnecessary “it”

L62 Should read “suggest”

L70 Replace “it as suggested” with either “It has been suggested” or “It is suggested”

Materials and Methods: Is it possible to explain/justify the sequence and time line? And, how can an arena be novel twice (once in MDTB and second in temperament assessment)?

Please justify the use of the net as predator simulation when they are caught like this on a regular basis. Couldn’t familiarity with the net effect the results at least for some animals?

L203 please add “the” before “animals” as you are referring to specific animals

L261 What is S1? Did you mean T1?

Discussion: L729 “do” should be “does”

L730 “resilient” should be “resilience”

Author Response

RESPONSE TO REVIEWER 1.

Temperament, plasticity and emotions in defensive behaviour of paca

Overall this paper is well written with lengthy introduction and descriptions of all the methods used. That said, the methods and their timing are complex and I must say a bit difficult to follow (with a few acronyms to remember for people not used to them).

Nogueira et al.: Thank you very much for your helpful comments and suggestions. We revised the ms trying to better explain the methods.

Specific comments include:

Simple summary: this summary is for lay persons so terms such as individuals’ plasticity and individuals’ affective states may not be clear.

Nogueira et al.: Yes, you are right. Instead individuals’ plasticity, we replaced to ‘individuals’ behavioural characteristics’ and ‘individuals’ emotions’

Abstract: Please put abbreviations in brackets after full term (e.g. JBT and MDTB).

Nogueira et al.: We add brackets in our abbreviations

Also, as in other places use present tense when talking about what the results ‘indicate’ (L28)

Nogueira et al.: “As commonly practiced, we reported our findings in the past tense to indicate that they arose from this specific experiment and are not definitive statements of fact about paca behaviour in general, as might be implied by use of the present tense. We therefore prefer to keep the past tense for reporting.”

“cross time” probably should be “across time”

Nogueira et al.: Yes, we corrected to “across time”

L32 Please delete “(to go)” as it just complicates the meaning

Nogueira et al.: Yes, we deleted “to go”.

Introduction: L60 an unnecessary “it”

Nogueira et al.: Yes, thank you. We deleted the “it”.

L62 Should read “suggest”

Nogueira et al.: Yes, we corrected replacing “suggested” to suggest.

L70 Replace “it as suggested” with either “It has been suggested” or “It is suggested”

Nogueira et al.: Yes, thank you. We replaced this with “It has been suggested”

Materials and Methods: Is it possible to explain/justify the sequence and time line?

Nogueira et al.: Yes, of course. We add such explanation as follows:

The timeline was designed to evaluate the individuals’ behavioural responses before and after the Modified Defence Test Battery-MDTB. Initially, we collected faecal samples to determine basal faecal glucocorticoid metabolite concentration before any interference with the animals (no stress induced) and at the end of the experiment to evaluate the increase of glucocorticoid metabolite concentration after the Modified Defence Test Battery-MDTB (after stress induced). To assess temperament across-time and contexts, we carried out tests in different situations and separated by at least 60 days. To evaluate affective states, we used a judgement bias test (JBT) which required initial training of animals to respond to cues by showing go or no-go responses, followed by testing on ambiguous cues before and after defensive behaviour to evaluate the effects on animals of stress induced by the Modified Defence Test Battery (see below).

And, how can an arena be novel twice (once in MDTB and second in temperament assessment)?

Nogueira et al.: We used two different environments, one for MDTB and the other for temperament tests. This was explained in our original version - “…To evaluate individuals’ defensive behaviour (details below), a test arena was built, following Nogueira et al. (2017). The arena (Figure 1B) had a corridor shape on a dirt floor with only a few occasional clumps of grass, measuring 15.0 m in length and 4.0 m in width, surrounded by a chain-link fence 1.5 m high to which zinc plates were fixed to both prevent visual contact with the outside environment and the animal’s escape. At the centre of this enclosure there was a chain-link wall 1.5 m high dividing the test arena into two parallel corridors measuring 2.0 m in width each…” – And to access animals temperament to novelty was explained in page 5, lines 215 to 219 in original version as follows- “For this test, an open field test arena square shape measuring ~10 m2 (Figure 1C) was used. Each individual was transported using a wood cage (0.6 m length x 0.4 m width x 0.3 m height) from its home pen to the open-field test arena.” The Figure 1, shows both arenas that were built in different shape and dimensions. 

Please justify the use of the net as predator simulation when they are caught like this on a regular basis. Couldn’t familiarity with the net effect the results at least for some animals?

Nogueira et al.: Taking in account the reaction of animals during this practice (panting, running and showing piloerection), we believe they do not get used to this procedure. We emphasize this in the new version as following: “This procedure would equate to a typical husbandry or predator event that pacas usually respond to by running away, painting and showing piloerection. The animals had never experienced true predation, although they did experience human predator-like cues (sensu Mason et al., 2013) when they were caught to be weighed every two months, and for occasional veterinary evaluation in case of injuries or sickness. As the animals always attempted to avoid being caught, we chose to use the capture net to stimulate the expression of defensive behavioural patterns, such as escape, during the chase and forced contact tests.

L203 please add “the” before “animals” as you are referring to specific animals

Nogueira et al.: Yes, thank you. We added “the”.

L261 What is S1? Did you mean T1?

Nogueira et al.: No, I am sorry, S1 is a supplementary file describing the adjectives used in this ms:(Supplementary 1. List of adjectives definitions used to evaluate the paca’s temperament traits). However, we had  some problem during submission that  do not upload this material as supplementary material. We decided to add as a new table (Table 1) in this version.

Discussion: L729 “do” should be “does”

Nogueira et al.: Yes, thank you. We correct it.

L730 “resilient” should be “resilience”

Nogueira et al.: Yes, thank you. We correct it.

Reviewer 2 Report

This study is well designed and written.

I have only one concern about the test animals.

The test animals were all born and bred in captivity (line 135). They are not caught from the wild population with heavy hunting practice. Therefore, the inferences made from the tests in this study might not be suitable to apply for the wild population. For example, in discussion (lines 675-688), the behavioral plasticity shown in this study, that is, rapidly change its behaviour from a very distressed condition to a more positive condition, might be simply that the test animals are born and bred in captivity. I suggest that authors take this concern into consideration in the discussion section.

Minor suggestions:

  1. Figure 1: please use the test abbreviation to replace the A-I on the timeline for the reader’s convenience.
  2. Table 2: in the note, rs > 0.7 might be better to use rs > 0.738 (the value for significance level at P =0.05 two-tailed).
  3. Table 4: This table demonstrates the measurement values not the correlation values.The table title should be changed. Additionally, in the text (lines 494, 506), the tests were done separately for roaming and raising forelegs; however, in the table, the two values were combined together as EXP3. Please show each instead combined one.

Author Response

RESPONSE TO REVIEWER 2.

This study is well designed and written.

I have only one concern about the test animals.

The test animals were all born and bred in captivity (line 135). They are not caught from the wild population with heavy hunting practice. Therefore, the inferences made from the tests in this study might not be suitable to apply for the wild population. For example, in discussion (lines 675-688), the behavioural plasticity shown in this study, that is, rapidly change its behaviour from a very distressed condition to a more positive condition, might be simply that the test animals are born and bred in captivity. I suggest that authors take this concern into consideration in the discussion section.

 Nogueira et al.: Thank you for your comments. Indeed, it is possible to have differences between captive born pacas and free range ones regarding their behaviours and experience with humans, which could also explain our results. We add a sentence considering your comment regarding differences in behaviour of captive born pacas and free range animals as follows-

“It is also important to consider that such rapid changes in behaviour from a very distressed state to a more positive state might be more common in captive individuals and relatively small individuals born and raised in captivity than under natural conditions. However, it is very difficult or even impossible to study plasticity of defensive behaviour of free-ranging pacas. Thus, it would be interesting to carry out further studies on larger samples of captive pacas in order to confirm the results described here.

Minor suggestions:

  1. Figure 1: please use the test abbreviation to replace the A-I on the timeline for the reader’s convenience.

Nogueira et al.: In Figure 1, A-I is just the original pen where animals use to live. However, we added to Figure 1 the same timeline terms referred to in Figure 2. We hope it is clearer this way. In this version the legend is as follows, “Diagrams of the pacas’ home pen (A); arena used for Modified Defence Test Battery (MDTB) (B) and open field used during temperament assessment (novel environment test) (C).”.

  1. Table 2: in the note, rs > 0.7 might be better to use rs > 0.738 (the value for significance level at P =0.05 two-tailed).

Nogueira et al.: In the revised version of the Methods we included the information that  we followed the procedures described by Feaver et al (1986) as follows: “Further analysis only involved those adjectives that showed inter-observer coefficients of concordance greater than or equal to 0.70, following the procedures described by Feaver, Mendl, & Bateson (1986).” We also included the P values (P<0.05 or P>0.05) in Table 2.

  1. Table 4: This table demonstrates the measurement values not the correlation values.The table title should be changed. Additionally, in the text (lines 494, 506), the tests were done separately for roaming and raising forelegs; however, in the table, the two values were combined together as EXP3. Please show each instead combined one.

Nogueira et al.: Thanks for pointing out this problem. Actually the values showed in Table 4 are the rSpearman values. We included this information in the title as follows:

“Table 4. Spearman coefficient of correlations (rSpearman values) among the mean ratings of selected adjectives.”

We added to the note the following information: *Bold rSpearman values represent rs > 0.70 with significance P<0.05 that were used to combine the ‘restless’ behavioural trait dimension.

Nogueira et al.: In the revised Table 5, we do prefer to show the joint analysis of the time spent (s) on these two behavioural patterns (roaming + raising forelegs) because both patterns belong to the same behavioural category (exploratory behaviour). We follow the same reasoning when analyzing the time spent (s) on trying to escape acts (running + climbing). Therefore, to show the results in a homogeneous way, in the revised version we showed the results of the correlation analyses between the individuals’ ‘restless’ scores and the time pacas spent on exploratory patterns together (roaming + raising forelegs) as well. Additionally, in the revised version we included the results of the Spearman correlation tests between the individual’s ‘restless’ scores and the time they spent on trying to escape (running + climbing) during both the novel-environment and the foraging-eating tests.

Nogueira et al.: Thank you very much for your helpful comments and suggestions.

Reviewer 3 Report

In this interesting and comprehensive study, the authors examined whether responses shown during temperament tests predicted behavior of pacas during the modified defense test battery and whether levels of glucocorticoid metabolites differed before and after the test battery. The results are interpreted in the context of conservation and vulnerabilities due to potential overhunting of the study species. My comments are listed by section below.

Simple Summary:

This section needs a description of the results. It moves directly from a description of methods (lines 16-18) to what results might mean (lines 18-23), without describing the results. Also, I suggest breaking up the very long sentence running from line 18 to line 23 to increase clarity.

Abstract:

Line 32: English editing required for the sentence beginning, “Restless animals were correlated”.

General comment: Note that the citations in the text do not follow journal guidelines (they should be numbered and within brackets, e.g., [1,2]). The reference section also needs reformatting - currently, references are listed alphabetically, and they should be listed by number, in order of occurrence in the text.

Introduction:

Line 60: delete “it”

Line 99: I am not sure “Transposing “ is the correct word here.

Lines102-121: This paragraph could be shortened by omitting some details, such as the 30 sec length of temperament tests and that tests occurred over 180 days. Also, I suggest reworking the paragraph, so that it is not simply a long list of things measured.

Lines 122-126: I like that predictions are included here, but this section would be strengthened by providing the bases for your predictions. For example, on what published results (rodents or other mammals) did you base your prediction that bolder pacas would be less affected by the modified defense test battery?

Materials and Methods:

Line 134: Was there a reason for only studying male pacas?

Figure 1 is very helpful. For consistency, I suggest providing the measurements for Part B, as done in Parts A and C (I know they are provided in the text, but I would include them in the figure).

Figure 2 also is very helpful.

Paragraph beginning on line 215: I might have missed it, but what was the substrate for the open field test arena and was it cleaned between pacas (this question also applies to the antipredator test since it also was conducted in the open field arena)?

General question: Were keepers and research assistants working with the pacas during testing males or females? Experimenter sex has been shown to affect behavior and physiology of rodents (e.g., Sorge, R.E. et al. Olfactory exposure to males, including men, causes stress and related analgesia in rodents. Nature Methods, 2014, 11, 629–632) as well as other mammals, such as dogs and elephants. Also, were keepers and research assistants familiar to the individual pacas tested? If so, how familiar? Familiarity also influences human-rodent interactions (e.g., Davis, H.; Taylor, A.A.; Norris, C. Preference for familiar humans by rats. Psychonom. Bull. Rev. 1997, 4, 118–120).

Section 2.7: I do not have the expertise needed to evaluate the methods of collecting, storing, and analyzing fecal glucocorticoid metabolites.

Line 436: How were flight initiation distance and flight speed measured from videos?

Results:

Line 472: “curious” is listed twice – change one to “anxious”?

Table 1 title: I suggest reminding readers of the three challenge tests by including their names in parentheses after mentioning inter-challenge tests (novel object, novel environment, and antipredator). There are many different tests described in the paper and this will help.

Figure 3: I would place the labels for individual pacas (e.g., Paca 1, Paca 2, etc.) either directly below each bar (for Pacas 1-4, and 6) or directly above each bar (Pacas 5,7 and 8). The current labels overlap some of the bars (for Pacas 1-4, and 6).

Table 3 should become Table 1 and it should be moved to Materials and Methods where it is first described in Section 2.6. Material relevant to methodology is presented in this table (definitions for behavioral categories); it does not contain results.

Table 4 title: “The relationship of temperament” seems too vague. I suggest being more specific and stating something like: “The relationship between restless temperament scores”. Also, regarding the footnotes, are the values for Abnormal behavior and Freezing also in time spent (s)? Continuing with the footnotes, units are needed here for flight speed and flight initiation distance.

Line 555: Change 90% to 80%?

Discussion:

Although study limitations are briefly mentioned in different paragraphs (e.g., line 633 and line 729), including a more complete list in a single paragraph might work better.

Line 646: Change “researches” to “reserves”.

Line 679 requires editing (“this may be occur”), as does line 685 (“the latter case possibly happen”).

Line 730: Change “resilient” to “resilience”.

Although possibly not relevant to this study, temperament tests/behavior evaluations of dogs in shelters have come under a lot of scrutiny in the last few years, especially those tests in the battery that are used by some shelters to screen dogs for adoption versus euthanasia. For example, the predictive ability of some tests (e.g., tests for resource guarding) has been shown to be relatively low - only about 50% of the time the behavior displayed during testing at the shelter is later shown in an adoptive home (e.g., Patronek, G.J.; Bradley, J. No better than flipping a coin: Reconsidering canine behavior evaluations in animal shelters. J. Vet. Behav. 2016, 15, 66–77. Patronek, G.J.; Bradley, J.; Arps, E. What is the evidence for reliability and validity of behavior evaluations for shelter dogs? A prequel to “No better than flipping a coin”. J. Vet. Behav. 2019, 31, 43–58). Have temperament tests in rodents been evaluated for measures such as sensitivity, specificity, false positives, etc.? (I realize that such measures require larger sample sizes than in the present study). Maybe somewhere in the Discussion the predictive abilities/usefulness/potential problems associated with rodent temperament tests in general should be covered?

Research articles in this journal require a separate Conclusions section, but I did not see one.

Paper on rodents that might be of interest:

Brehm et al. 2020. Effects of trap confinement on personality measurements in two terrestrial rodents PloS ONE 15(1): e0221136.

I enjoyed reading this manuscript and hope the authors find my suggestions helpful.

Author Response

RESPONSE TO REVIEWER 3

In this interesting and comprehensive study, the authors examined whether responses shown during temperament tests predicted behavior of pacas during the modified defense test battery and whether levels of glucocorticoid metabolites differed before and after the test battery. The results are interpreted in the context of conservation and vulnerabilities due to potential overhunting of the study species. My comments are listed by section below.

Simple Summary:

This section needs a description of the results. It moves directly from a description of methods (lines 16-18) to what results might mean (lines 18-23), without describing the results. Also, I suggest breaking up the very long sentence running from line 18 to line 23 to increase clarity.

Nogueira et al.: Yes, thank you. We reworded this section adding results and breaking up the final sentence as follows: “…Our results showed that paca with a ‘restless’ temperament performed more abnormal behaviour and less exploratory behaviour in a test of defensive behaviour which elevations in faecal glucocorticoid metabolites indicated to be stressful. Plasticity in defensive behaviour was inferred from changes in behavioural responses and apparently rapid adaptation to challenge in the different levels of risk. Our results suggest that individual differences and consistency of behavioural responses displayed by paca toward challenges may reflect a generally flexible and successful defensive behavioural response that underpins the paca’s survival despite the threat of overhunting throughout its range.” 

Abstract:

Line 32: English editing required for the sentence beginning, “Restless animals were correlated”.

Nogueira et al.: Yes, thank you. We reworded this sentence as follows- Restless animals were more likely to show a greater proportion of positive responses to an ambiguous cue during JBT after the MDTB.

General comment: Note that the citations in the text do not follow journal guidelines (they should be numbered and within brackets, e.g., [1,2]). The reference section also needs reformatting - currently, references are listed alphabetically, and they should be listed by number, in order of occurrence in the text.

Nogueira et al.: Yes, you are right, however, the journal allows free reference format up to when the ms is accepted. In this version we already use the proper format, thank you for noticing this.

Introduction:

Line 60: delete “it”

Nogueira et al.: Yes, thank you. We deleted the “it”.

Line 99: I am not sure “Transposing “ is the correct word here.

Nogueira et al.: We replaced the word “Transposing” by “Considering”.(See Line 140 of new version)

Lines102-121: This paragraph could be shortened by omitting some details, such as the 30 sec length of temperament tests and that tests occurred over 180 days. Also, I suggest reworking the paragraph, so that it is not simply a long list of things measured.

Nogueira et al.: Yes, I understand your point, our study indeed has measured several things and we feel that it is important to describe those things in some detail to enable replication by others and make the theoretical background clear. However, we have tried to cut some information that is not essential. The paragraph is now as follows below:

To investigate these issues, we measured the responses of captive pacas to three short challenge tests to assess indicators of their temperament. Additionally,, we analysed their defensive responses using the modified defence test battery (MDTB: novel environment; chase; forced contact with human; foraging-eating in novel environment) (Blanchard, Griebel, & Blanchard, 2003), and investigated links between the emerging temperament measures and response to simulated hunting. We also measured faecal glucocorticoid metabolites prior to and following the MDTB to determine whether temperament measures were associated with physiological stress responses to the test events   (Charmandari, Tsigos, & Chrousos, 2004; Drent, Oers, & Noordwijk, 2003; Gunnar & Quevedo, 2007). In addition, we trained pacas on a judgement bias test (JBT) (Harding, Paul, & Mendl, 2004; Mendl, Burman, & Parker, 2009), which involves assessing responses to ambiguous stimuli that predict potentially positive or negative outcomes. This test has generally been used as a marker of animal affective states following the hypothesis, based on human psychology studies (e.g. MacLeod et al., 2002; Paul, Harding, & Mendl, 2005), that individuals in a more negative state will be more likely to treat ambiguous stimuli ‘pessimistically’ as predicting a negative outcome. Here we also measured speed of learning of the discrimination task on which the JBT is based as an indication of behavioural plasticity, as well as responses to ambiguous stimuli during JBTs carried out before and after exposure to the MDTB.”

Lines 122-126: I like that predictions are included here, but this section would be strengthened by providing the bases for your predictions. For example, on what published results (rodents or other mammals) did you base your prediction that bolder pacas would be less affected by the modified defense test battery?

Nogueira et al.: Yes, thank you. We reworded this paragraph including some references as suggested as follows,

“We predicted that we would find stable variation in our temperament measures amongst pacas, as observed in other captive wild mammals (e.g. Pecari tajacu and Tayassu pecari, Nogueira et al. 2015a). We also predicted that individuals that were bolder and less disturbed in challenge tests would also be less affected by the MDTB, show a smaller physiological stress response, and make less ‘pessimistic’ decisions in the JBT, because links between behavioural and physiological responses to challenge have been observed in other rodents (e.g. Matzel &Sauce 2017; Cavigeli 2018; Lima et al. 2019). We also expected to see plasticity of responding during the duration of the MDTB, as pacas exhibit behavioural flexibility (Figueroa-de León et al. 2016; van Vliet and Nasi 2018).”

Materials and Methods:

Line 134: Was there a reason for only studying male pacas?

Nogueira et al.: Yes, We avoid using females to avoid bias caused by hormone changes across the ovulatory cicle as the whole study took over 180 days to assess all behavioural responses. We added this information in this new version (see lines 171-172 of the new version.

Figure 1 is very helpful. For consistency, I suggest providing the measurements for Part B, as done in Parts A and C (I know they are provided in the text, but I would include them in the figure).

Nogueira et al.: Yes, thank you for this suggestion. We agree and added the measurements for Part B.

Figure 2 also is very helpful.

Nogueira et al.: Thank you very much.

Paragraph beginning on line 215: I might have missed it, but what was the substrate for the open field test arena and was it cleaned between pacas (this question also applies to the antipredator test since it also was conducted in the open field arena)?

Nogueira et al.: Thank you very much, you are right I forgot to add these important pieces of information. The open field test arena was built on dirt floor and we cleaned the floor before each test (either novel environment or anti-predator tests). Additionally, all tested objects (net and ball) were also cleaned during the intervals. We add this information as follows:

 “After the end of the test, the keeper collected the ball, cleaned it with a damp cloth soaked with a substance of 70% ethanol, 1% acetic acid (see McGuire et al. 2012)…”

 “For this test, a square open field test arena square on a dirt floor measuring ~10 m2 (Figure 1C) was used.”

 “…During this interval, the keeper sprayed the arena with a solution composed of 70% ethanol, 1% acetic acid to mask the smell of the previous animal and any pheromonal signals they may have left, as recommended by McGuire et al. (2012).”

 “…During the interval, the keeper sprayed the arena with the same solution described above following McGuire et al’s (2012) recommendation”

General question: Were keepers and research assistants working with the pacas during testing males or females? Experimenter sex has been shown to affect behavior and physiology of rodents (e.g., Sorge, R.E. et al. Olfactory exposure to males, including men, causes stress and related analgesia in rodents. Nature Methods201411, 629–632) as well as other mammals, such as dogs and elephants. Also, were keepers and research assistants familiar to the individual pacas tested? If so, how familiar? Familiarity also influences human-rodent interactions (e.g., Davis, H.; Taylor, A.A.; Norris, C. Preference for familiar humans by rats. Psychonom. Bull. Rev. 19974, 118–120).

Nogueira et al.: Thank you for your comment, the keeper was male and research assistant was female. The keeper was very familiar to animals, he used to treat them daily and research assistant was also familiar to them, and had habituated them for a previous study. We add this information in new version as follows:

 Both the keeper (male) and research assistant (female) were familiar to the animals. The keeper interacted with the animals daily and the research assistant habituated the animals to her presence in a previous behavioural study so as to minimise any effects of sex (Sorge et al. 2014) and familiarity (Davis et al. 1997) of humans on experimental results.”

Section 2.7: I do not have the expertise needed to evaluate the methods of collecting, storing, and analyzing fecal glucocorticoid metabolites.

Nogueira et al.: Thank you about your concern, however, our co-author Prof. Duarte has expertise in this field and we followed all proper methods.

Line 436: How were flight initiation distance and flight speed measured from videos?

Nogueira et al.: Thank you for noticing that, we add this procedure in the new version as follows:

Watching the video-recorded images, a single observer determined the individual’s flight initiation distance (FID) and flight speed (FS). The FID was calculated from the moment the animal started to react to the capture net. The observer used the chalk lines made on the test arena floor as a marker to estimate this distance using the metric system. The FS was also estimated with the chalk lines and a simple chronometer to measure the distance covered by the animal per unit time.. All video footage analyses were made by the same observer to avoid measurement bias. Thereafter, we determined the means of FID and FS from the three chases in each test. Both measures were scored as zero for the animals that did not run away from the threat. 

Results:

Line 472: “curious” is listed twice – change one to “anxious”?

Nogueira et al.: yes, thank you very much. It is supposed to be “anxious”. We have corrected this mistake.

Table 1 title: I suggest reminding readers of the three challenge tests by including their names in parentheses after mentioning inter-challenge tests (novel object, novel environment, and antipredator). There are many different tests described in the paper and this will help.

Nogueira et al.: Yes, you are right. We added this information as follows:

: “Table 3. Inter-observer and inter-challenge tests (novel object, novel environment, and antipredator) using Kendall’s concordance coefficients (W) of individual behavioural distinctiveness traits.”

Figure 3: I would place the labels for individual pacas (e.g., Paca 1, Paca 2, etc.) either directly below each bar (for Pacas 1-4, and 6) or directly above each bar (Pacas 5,7 and 8). The current labels overlap some of the bars (for Pacas 1-4, and 6).

Nogueira et al.: In the revised version we fixed it. Thank you.

Table 3 should become Table 1 and it should be moved to Materials and Methods where it is first described in Section 2.6. Material relevant to methodology is presented in this table (definitions for behavioral categories); it does not contain results.

Nogueira et al.: Thank you, we accepted your suggestion moving the Table to the appropriate position.

Table 4 title: “The relationship of temperament” seems too vague. I suggest being more specific and stating something like: “The relationship between restless temperament scores”. Also, regarding the footnotes, are the values for Abnormal behavior and Freezing also in time spent (s)? Continuing with the footnotes, units are needed here for flight speed and flight initiation distance.

Nogueira et al.: We accepted your suggestion and changed the Table title. We included the missing units as well. Thank you. Due to reordering of tables, we numbered this as Table 5 in the revised version.

Line 555: Change 90% to 80%?

Nogueira et al.: Thank you. We changed it.

Discussion:

Although study limitations are briefly mentioned in different paragraphs (e.g., line 633 and line 729), including a more complete list in a single paragraph might work better.

Nogueira et al.: Yes, we joined our limitation in one paragraph as follows:

Despite attempts to design a robust experiment within the constraints of our lab environment, our study inevitably has limitations that prevent us from drawing firm conclusions about the paca’s resilience to over-hunting by humans. These include the small sample size, the use of qualitatively different sound cues in the judgement bias tests that may have resulted in all animals treating the ‘ambiguous’ cue as being more similar to the positive one, and the fact that the predictive reliability of the temperament tests used here still requires further validation (e.g. Patronek & Bradley 2016; Patronek et al. 2019).  Thus, further studies must be done to improve and confirm our conclusions.

Line 646: Change “researches” to “reserves”.

Nogueira et al.: Yes, thank you very much it was typo problem. We corrected it (see line 811 of new version).

Line 679 requires editing (“this may be occur”), as does line 685 (“the latter case possibly happen”).

Nogueira et al.: Yes, thank you. We edit it as follows:

 “…This possibly occur in situations of high hunting pressure…”

: “…In relatively low hunting areas, the latter case may occur, allowing the paca to disperse and…”

Line 730: Change “resilient” to “resilience”.

Nogueira et al.: Yes, thank you we correct this.

Although possibly not relevant to this study, temperament tests/behavior evaluations of dogs in shelters have come under a lot of scrutiny in the last few years, especially those tests in the battery that are used by some shelters to screen dogs for adoption versus euthanasia. For example, the predictive ability of some tests (e.g., tests for resource guarding) has been shown to be relatively low - only about 50% of the time the behavior displayed during testing at the shelter is later shown in an adoptive home (e.g., Patronek, G.J.; Bradley, J. No better than flipping a coin: Reconsidering canine behavior evaluations in animal shelters. J. Vet. Behav. 201615, 66–77. Patronek, G.J.; Bradley, J.; Arps, E. What is the evidence for reliability and validity of behavior evaluations for shelter dogs? A prequel to “No better than flipping a coin”. J. Vet. Behav. 201931, 43–58). Have temperament tests in rodents been evaluated for measures such as sensitivity, specificity, false positives, etc.? (I realize that such measures require larger sample sizes than in the present study). Maybe somewhere in the Discussion the predictive abilities/usefulness/potential problems associated with rodent temperament tests in general should be covered?

Nogueira et al.: Thank you for this useful point. We now mention the potential limitations of temperament tests and the need to further investigate how reliable they are in terms of prediction, adding the two references you have provided (see answer to earlier comment).

Research articles in this journal require a separate Conclusions section, but I did not see one.

Nogueira et al.: Yes, thank you. We added a separate conclusion in the present version as follows:

Lines (910-930): The pacas studied herein showed cross-time and context stability in temperament dimension labelled ‘restless’. Individual ‘restless’ scores predicted responses to novelty, although not to simulated chasing and capture by humans in a separate Modified Defence Test Battery-MDTB. More ‘restless’ animals showed a greater proportion of positive responses (to go) to an ambiguous cue during JBT after the MDTB. Plasticity in defensive behaviour was inferred from changes in behavioural responses and apparently rapid adaptation to challenge in the different phases of the MDTB. The results indicate that both temperament and behavioural plasticity may play a role in influencing paca responses to risky situations. Therefore, our results allow us to suggest that individual differences and consistency of behavioural responses displayed by paca toward the MDTB challenges, together with the species’ ability to modulate their responses according to risk levels and individual variability in learning the negative conditioned stimulus (CS-), may reflect a generally flexible and successful defensive behavioural response that underpins the paca’s survival despite the threat of overhunting throughout its range.

Paper on rodents that might be of interest:

Brehm et al. 2020. Effects of trap confinement on personality measurements in two terrestrial rodents PloS ONE 15(1): e0221136.

I enjoyed reading this manuscript and hope the authors find my suggestions helpful.

Nogueira et al.: Thank you very much for your helpful comments and suggestions.